# Dementia- and mild cognitive impairment-inclusive exercise: Perceptions, experiences, and needs of community exercise providers

**Lauren E. Bechard**[1]*, **Aidan McDougall**[1], **Cheyenne Mitchell**[1], **Kayla Regan**[1], **Maximillian Bergelt**[1], **Sherry Dupuis**[2], **Lora Giangregorio**[1], **Shannon Freeman**[3], **Laura E. Middleton**[1]

**1** Department of Kinesiology, Faculty of Applied Health Sciences, University of Waterloo, Waterloo, Waterloo, Ontario, Canada, **2** Department of Recreation and Leisure Studies, Faculty of Applied Health Sciences, University of Waterloo, Waterloo, Waterloo, Ontario, Canada, **3** University of Northern British Columbia School of Nursing, Prince George, British Columbia, Canada

* lauren.bechard@uwaterloo.ca

**Data Availability Statement:** All relevant data are within the paper and its Supporting Information files.

## Abstract

### Purpose

For persons who are at risk for, or living with, dementia exercise is recommended, yet many become or remain inactive. Exercise providers play a vital role in promoting and facilitating exercise in these groups by recognizing and being responsive to the needs of persons with mild cognitive impairment (MCI) or dementia in exercise programming. The objective of this study was to explore the experiences, perceptions, and needs of community exercise providers regarding dementia.

### Materials & methods

Five focus groups were held with community exercise providers (n = 30) who deliver exercise to older adults (≥55 years) in municipal, non-profit, for profit, or academic settings.

### Results

Three themes were developed: (1) *Unique experiences and diverse perceptions*: suggests unique personal experiences with MCI and dementia inform distinct perceptions of dementia; (2) *Dementia-Inclusive Practices*: *learning as you go and adapting for the individual*: reflects exercise providers' approaches to recognizing and accommodating individuals' unique abilities and preferences; (3) *Training and Best Practices*, *with Flexibility*: identifies exercise providers' desires for MCI- and dementia-specific knowledge and training strategies, which need to recognize dementia heterogeneity between and within persons over time.

### Conclusions

These findings highlight a willingness of exercise providers to support dementia-inclusive exercise, but recognize they have minimal training and lack educational resources to do so.

**Funding:** This work was supported by: (1) the Royal Bank of Canada Undergraduate Research Fellowship (CM) - https://www.rbc.com/dms/enterprise/scholarships.html (2) the Hallman Undergraduate Research Fellowship (AM) - https://uwaterloo.ca/applied-health-sciences/faculty-staff/research-funding/hallman-undergraduate-research-fellowship (3) the Canadian Institutes for Health Research (grant #01881-000) (LM) - https://webapps.cihr-irsc.gc.ca/decisions/p/project_details.html?appId=378492&lang=en (4) the University of Waterloo Chronic Disease Prevention Initiative (LM) - https://uwaterloo.ca/propel/ The sponsors / funders did not play any role in the study design, data collection and analysis, decision to publish, or preparation of the manuscript.

**Competing interests:** The authors have declared that no competing interests exist.

Formal training resources may enhance exercise accessibility and participation for persons with MCI or dementia.

## Introduction

Dementia is one of the most pressing social challenges we will face over the coming decades, with global prevalence anticipated to almost double every two decades from 46.8M people affected in 2015 to 131.5M people by 2050 [1]. The global economic burden of dementia is expected to surpass US$2T as soon as 2030 [1]. Aside from the direct health effects of dementia to the individual, it also affects one's ability to participate in meaningful daily live activities and can disrupt important social relationships. Presently, there are no disease-modifying therapies available to treat or cure dementia. Current pharmacological treatments may slow cognitive decline, but in the scale of months not years [2].

Exercise, which is physical activity done with the intent to improve one's health and function [3], is a non-pharmacological strategy to reduce the risk of dementia among persons with mild cognitive impairment (MCI) and can improve well-being of persons with either MCI or dementia. Exercise improves fitness, balance, and mobility among persons with MCI or dementia [4–8]. Functional abilities also improve with exercise among persons with dementia, which may contribute to reduced care partner stress [4]. Randomized clinical trials of exercise also show improved cognition among persons with MCI and dementia [9–12], such that exercise is currently recommended as part of the clinical management of MCI [13]. Cognitive benefits among persons with MCI and dementia (especially Alzheimer's disease) were observed after moderate intensity exercise of various types including aerobic exercise, resistance training, and tai chi [9–12, 14].

Functional and cognitive benefits are often the focus of clinical trials of exercise in dementia and MCI, but exercise may have broader benefits to participant's emotional and social well-being. Persons with MCI or dementia report that exercise is an opportunity for social engagement and to demonstrate continued capacity [15–18]. Long-time exercisers, in particular, may view exercise as a personally meaningful activity [19–21], which can be an opportunity to maintain their identity [22, 23].

The United Nations Convention on the Rights of Persons with Disabilities specifically recognizes the right of persons living with disability (including persons with MCI or dementia) to "*participate in cultural life, recreation, leisure, and sport*" [24]. As such, communities and organizations offering exercise programs have a duty to make exercise programs accessible to persons living with MCI or dementia. The limited data available on exercise levels amongst persons with MCI or dementia suggest they are less active than older adults without cognitive impairment, and that few meet physical activity and fitness levels recommended for the maintenance of health and function [25, 26].

Trends of low physical activity and fitness amongst persons with MCI or dementia may be due to several types of barriers they face regarding exercise participation. For one, persons living with MCI or dementia are typically older, and as such must contend with perceived age-related barriers to exercise participation such as fears of falling, increased disease burden, and limiting beliefs about the potential for experiencing the benefits of exercise at advanced ages [27]. Compounding these age-related barriers to physical activity participation, persons with MCI or dementia also face participation barriers related to MCI and dementia symptoms, such as memory impairment, difficulty wayfinding, difficulty with goal-directed behaviours,

apathy, balance and gait impairment. These symptoms can make accessing individual and group exercise opportunities more difficult [19, 22, 28, 29]. Social factors faced by persons living with MCI or dementia such as stigma and lack of understanding about the abilities of these individuals by both the general public and exercise professionals may create another layer of barriers to participation [19, 20].

Guidance from a knowledgeable professional is a facilitator to exercise participation among older adults with and without cognitive impairment [27, 30], but reliance on exercise guidance from healthcare professionals (e.g. physicians, nurses, physiotherapists) may not be sustainable due to cost and accessibility. Community exercise providers constitute a diverse group (e.g., fitness instructors, personal trainers, certified exercise professionals, kinesiologists) that are uniquely poised to mitigate the individual and social barriers to exercise among persons with MCI or dementia. To do so, the current perspectives and experiences of community exercise providers must be understood to determine how exercise is already being provided to community-dwelling persons living with MCI or dementia. Yet, few studies have examined exercise providers' perspectives and experiences with exercise in MCI and dementia, and existing research focuses on health professionals rather than community exercise providers [30, 31]. Thus, the aim of the current study was to explore the experiences, perceptions, and needs of exercise providers in relation to supporting exercise participation among persons with MCI or dementia in community settings.

## Materials and methods

### Study activities

This qualitative study was conducted to explore the perspectives, experiences, and needs of community-exercise providers regarding exercise for people with MCI or dementia. Focus groups were held with community exercise providers with diverse professional and educational backgrounds and practice settings to elicit their beliefs about exercise in the context of MCI or dementia, experiences providing exercise to clients with MCI or dementia, and perceived needs to improve inclusivity and accessibility of community exercise programs for persons with MCI or dementia.

Exercise providers were recruited to participate in this study through community organizations offering exercise programs and services for older adults in and around an urban center in Southern Ontario, Canada. Study eligibility criteria were intentionally broad to capture a variety of perspectives; participants could be exercise providers delivering exercise of any format (e.g. personal training, group exercise) to persons aged 55 years or older at least once weekly. Potential participants were identified through outreach to community exercise organizations that had existing partnerships with the researchers as well as non-partner organizations, including for-profit, non-profit, municipal recreation, and academic centres. The study coordinator contacted organizations via email with study information to share with their staff, who could choose to participate by contacting the study coordinator. Where there were multiple participants from a single organization interested in participating, focus groups could be conducted in the community at their organization. Focus groups with participants from diverse organizations were conducted at a university research facility.

Prior to beginning study participation, all participants provided written, informed consent to participate in this research study and were given an opportunity to address any concerns related to participation with research personnel. Participants completed focus groups and demographic questionnaires in one study session held either at the research facilities of the investigators or in the community at the location of a participating organization between June to November 2017. During the study sessions, participants first completed a questionnaire

regarding themselves and their practice, followed by participation in a focus group discussion. The following data were collected from participants to characterize the study group: demographic information (age, gender, occupation), exercise qualifications, years of experience as an exercise provider, setting of exercise practice (e.g. clinic, community centre), location of practice (self-reported rural/urban), type of exercise training offered (individual, small group, exercise classes), and estimated age range of their clients. Participants also indicated whether any of their clients either: i) had known diagnoses of MCI or dementia; and/or (ii) demonstrated symptoms of cognitive impairment (e.g. difficulty remembering things, trouble focusing their attention or making decisions).

Focus groups were facilitated by a trained moderator (CM or NH) and an assistant moderator (MB or LM). Focus groups began with introductions of the research team and a brief description of study purpose, procedures, and guidelines for participation. The moderator opened the focus group discussion with an introductory question, answered by each participant. Following the opening question, discussion was facilitated in a semi-structured manner, referring to the focus group discussion guide as necessary (S1 File). The focus group discussion guide was intentionally designed to probe participants' experiences, perceptions, and needs concerning dementia and exercise among persons with dementia.

Both the moderator and assistant moderator took field notes for the purpose of debriefing and clarity. The assistant moderator recorded the session. The goal was to promote discussion until natural expiration. At the conclusion of each focus group, the assistant moderator provided a summary of the main points of the discussion and asked if participants had any clarifications or points to add. When no additional comments or clarifications were offered, participants were thanked for their contributions by the focus group facilitators and study participation was concluded.

## Data analysis

All focus group were audio-recorded and transcribed verbatim by trained research assistants. These transcripts from focus groups provided the data used for conducting inductive, qualitative thematic analysis in this study to identify common patterns in how the experiences, perceptions, and needs of exercise providers inform their ideas about and provision of exercise for older adults with cognitive impairment [32]. Thematic analysis was chosen as the analytical approach for this study due to its theoretical flexibility. Given that there is little research currently available on exercise providers experiences, perceptions, and needs in providing dementia-friendly exercise, this theoretical flexibility allows for an exploration of the data that is not framed by the researchers' preconceptions about what theoretical framework would be most appropriate. This flexibility aligns well with the inductive approach [32] taken to analysis in this study, in that coding and thematic organization were driven by the ideas and statements of participants, and not a pre-determined theoretical framework. As the focus of this study was on describing the experiences, perceptions, and needs of exercise providers, analysis was kept as close to the data as possible by using a semantic approach [32], such that the words and ideas of the participants were taken at face-value, rather than using a latent approach and interpreting the underlying meanings of exercise providers' experiences, perceptions, and experiences.

The process of thematic analysis followed in this study is described below. Transcription of interviews was performed by research assistants, and thorough reading and re-reading of all transcripts was undertaken by the authors involved in open coding (LB, AM) to familiarize themselves with the data. One author (LB) developed an initial coding structure through an iterative process, progressively refining the coding structure after open coding of sequential focus groups. In tandem with this process, coding with three secondary coders (CM, AM, MB) was

performed for a portion of a single transcript to establish a consensus coding structure to enhance the confirmability of results and reduce the influence of an individual researcher's biased perspective on data analysis [33]. Double-coding of transcripts using this coding structure was subsequently performed by two authors (LB, AM) to enhance confirmability, and the initial coding structure was continually refined throughout this process as analysis progressed. Discrepancies in coding between the two authors were discussed until a resolution was determined. After open coding was completed, an initial thematic organization developed by one author (LB) was presented to the research team for review along with supporting quotes to assess representativeness of the thematic groups. This process of reviewing and eliciting feedback from the broader research group not involved in coding on the organization and meaning of the thematic groups was conducted to enhance confirmability. Theme names and definitions were developed following this presentation and review, then re-presented to the research group for review with supporting quotations to assess theme representativeness. Following this process, final definitions and names of themes were developed by two of the authors (LB, LM) for reporting.

Trustworthiness of results was enhanced through several processes [34, 35]. Member checks were performed at the end of each focus group so that participants could verify interpretations of the moderating team. The credibility of findings was improved by frequent debriefing sessions with the research team to allow for peer scrutiny and the use of multiple coders during the analysis process to prevent a single investigator's perspectives from colouring the interpretation of the data. To support transferability and transparent interpretation of results, a detailed description of the study setting, participants, and eligibility criteria is reported in this paper. Finally, preservation of audio recordings, coded transcripts with annotations, and debriefing notes provide an audit trail.

## Research ethics

This study received ethics clearance through a University of Waterloo Research Ethics Committee (ORE #21362). All participants provided written informed consent prior to participating in the study. Participant confidentiality was maintained by de-identifying participants during the process of transcribing focus group recordings and questionnaire data. Numerical participant identifiers were used instead of names in transcripts and potentially identifying details provided during focus groups were anonymized. Only aggregate participant data from questionnaires are presented in the results section of this paper. Illustrative quotes included in the results section are de-identified and anonymized, but the focus group number and professional background of the participant being quoted are included to provide the reader with context for interpreting the results.

## Results

Five focus group sessions were held with a convenience sample totaling 31 participants, representing six organizations. Focus groups lasted between 62 minutes and 108 minutes (median of 72 minutes). In brief, most participants were female (90%, n = 28) and affiliated with public community centres (65%, n = 20) in urban settings (97%, n = 30). Participants had a variety of professional backgrounds but were most frequently a "personal trainer" (32%, n = 10), "exercise physiologist" (32%, n = 10), or held other administrative roles (e.g. program facilitator, supervisor) (42%, n = 13). A more detailed demographic and professional profile of participants is provided in Table 1.

Three themes relating to the perceptions, practices, and needs of exercise providers concerning inclusive exercise for persons with MCI or dementia were generated through the process of analysis [32]. The ordering of themes reported in this manuscript is done intentionally

**Table 1. Study participant profile.**

| Characteristic | Mean (range), or n (%) |
|---|---|
| **Age (years)** | **45 (26–76)** |
| **Years of Practice** | **9 (0–25)** |
| **Gender, female** | **28 (90%)** |
| **Organization type** | |
| Public community centres | 20 (65%) |
| Commercial fitness centres | 14 (45%) |
| Independent exercise providers | 3 (10%) |
| Academic fitness centres | 4 (13% |
| Other | 6 (19%) |
| **Professional Backgrounds**[a] | |
| Personal trainer | 10 (32%) |
| Registered kinesiologist | 5 (16%) |
| Fitness instructor | 5 (16%) |
| Exercise physiologist | 10 (32%) |
| Other (supervisor, program coordinator) | 13 (42%) |
| **Rural practice setting** | 1 (3%) |

[a] Professional backgrounds are not mutually exclusive, with exercise providers able to hold more than one exercise qualification.

to show how the personal and professional experiences of exercise providers were foundational to their perceptions of MCI and dementia, both in general and in relation to exercise provision. Exercise providers' perceptions then informed their current practices for exercise provision to persons with MCI or dementia, and the resources and education they felt were required to support the participation of persons with MCI or dementia in exercise.

## Theme 1—Unique experiences and diverse perceptions

This theme describes how the diverse personal and professional experiences of exercise providers have with persons with MCI or dementia inform their perceptions of MCI and dementia more broadly. Both the frequency and nature of personal and professional experiences with MCI and dementia contributed to forming these diverse perceptions.

Professional experiences, in particular, varied depending on the settings in which participants worked, ranging from private, for-profit fitness centers to specialized programs for persons with neurological conditions. Some exercise providers reported only occasionally encountering persons with MCI or dementia, whereas other exercise providers reported regular interactions encountering persons with dementia in their work. Exercise providers who reported extensive experiences with MCI and dementia in professional settings were more likely to recognize the diversity of abilities and experiences of persons with MCI or dementia.

*"I would say [my experience with dementia is] very limited, even professionally. I ran into some of the wellness folks that want to graduate into the group fitness so I have the odd participant."*

*(Focus Group 2, Group Fitness Supervisor)*

*"I'm a neurological supervisor here and I'm also a therapist in a neurological private clinic. Here, we deal a lot with individuals with varied forms of cognitive impairments, and outside*

*of here, **a majority of participants I deal with do have some form of dementia or other cognitive impairment of some kind**.*"

*(Focus Group 2, Neurological Exercise Supervisor)*

There were some participants, however, who had few interactions or only encountered persons with dementia in specific contexts in their professional experience, leading to more narrow perceptions of the abilities of persons with dementia or MCI. For example, one participant had primarily encountered acute situations of conflict with persons with dementia while working as hospital security:

*"The experience I've had with people with this [dementia] has been personal, prominently in a previous job I had working for security in a hospital. . . **I saw it [dementia], pretty much in . . . more or less acute situations**. So not being able to see any progression or anything like that. And uh, mostly just dealing with short term situations."*

*(Focus Group 4, Personal Trainer)*

In addition to diverse professional experiences with dementia, several exercise providers discussed their personal experiences with family members or friends with dementia. Some participants were very involved in the day-to-day activities of their family member or friend living with dementia, giving them familiarity with the reality of living with dementia. For example, one participant was a young carer for her grandmother with dementia. Other participants had a limited number of personal interactions with more distant friends or family members.

*"**My personal experience would be with my cousin' husband who's been diagnosed [with dementia]** . . . I've just seen a change in him, almost becoming child-like again . . . **wanting to do more physical things. Like dance–[he] loves to dance**! And, his whole married life, you couldn't get him on a dance floor. So, it's like he's reverted back. . . **and it's lovely to see.** Singing too!"*

*(Focus Group 4, Personal Trainer)*

*"**I've also had family connection; was a young carer for my grandmother**."*

*(Focus Group 5, Day Program Aid)*

During interviews, participants were asked broadly what came to mind when they thought of someone with MCI or dementia. Participants responding to this question had symptom-focused perceptions of MCI and dementia, and tended to describe cognitive and behavioural changes that can occur in MCI and dementia. When describing cognition, participants focused on impairments in memory, judgement, decision making, and communication abilities. Participants also described personality changes and agitation as common to MCI and dementia, but participants with more experience interacting with persons with MCI or dementia tended to more readily identify the diversity in symptoms and experiences that one can experience in MCI or dementia.

*"I would say **forgetfulness**. . . They've might have done the thing [exercise] a hundred times already but, you have to start back at the basics and explain it to them again, and repeat it, and show it to them again because **they just don't remember**."*

*(Focus Group 4, Personal Trainer)*

*"**Poor judgment** sometimes. . . maybe choosing to do something that we would consider dangerous for them, like getting on a bus and heading downtown because they used to come down for work and [they] don't realize they're not able to do that anymore."*

*(Focus Group 5, Day Program Aid)*

*"Their **personality really varies**. What you've got today isn't necessarily what you've got tomorrow or even ten minutes from now. And if you're lucky sometimes you can turn it around . . . I've found of lot of times I can see in their eyes what kind of day they're going to have."*

*(Focus Group 2, Fitness Instructor)*

*"I find that they get **really agitated at simple things** and even if you try to remain calm, they can heighten their aggression. . . I think it's because their [they're] **getting frustrated because they aren't comprehending what's going on**."*

*(Focus Group 2, Wellness Specialist)*

Based on the breadth of exposure to MCI and dementia that participants reported across both personal and professional settings, individual exercise providers viewed dementia through different lenses. Diverse experiences and perceptions influenced how participants approached exercise provision for persons with dementia. Some participants reported employing an individually-tailored approach for the persons with dementia which is the basis for the second theme identified during analysis.

### Theme 2—Dementia-inclusive practices: Learning as you go and adapting for the individual

This theme reflects how exercise providers participating in this study recognized and accommodated persons with MCI and dementia within their practice. This theme is comprised of two subthemes describing how exercise providers' perceptions of dementia influence their recognition and management of dementia in exercise settings: 2a) different ways of identifying persons with MCI or dementia in their programs; and 2b) developing *ad hoc* strategies to overcome barriers to exercise among persons with MCI or dementia and support inclusion.

Sub-theme 2a: Different ways of identifying persons with MCI or dementia in their programs. Exercise providers described a variety of formal and informal strategies for identifying persons with MCI and dementia they encountered in their practice. Some providers used detailed program intake assessments to characterize participants prior to exercise participation. In some cases, these assessments captured cognitive diagnoses either through explicit questions regarding health conditions, or occasionally through notes from referring family physicians. Exercise providers described these intake assessments being valuable for tailoring exercise programs to an individual's abilities and providing a motivational strategy by designing programs to support the achievement of personally-meaningful goals.

*"We're fortunate in, in our setting because we always start with an individual assessment of each individual who comes into our program. . . **We have an hour and a half to two-hour individual consultation with each individual who is referred to us**."*

*(Focus Group 3, Registered Kinesiologist)*

However, not all organizations had formal intake assessments that would capture a diagnosis of MCI or dementia. Even for organizations using formal intake assessments, exercise providers noted long-term participants may not have had discernable cognitive impairment at intake but may experience progressive cognitive decline over the duration of their participation in programs. For these reasons, exercise providers also relied on informal strategies to recognize cognitive decline and dementia in their clients. Informal strategies to identify dementia usually relied on observation of cognitive and behavioural symptoms that exercise providers associated with MCI or dementia (identified in Theme 1).

*"If they were diagnosed with something, that would come up in that initial pre-assessment with them. . . but there are some times where you kind of notice people over time like that, **you know it's becoming more and more apparent that there [is] an issue** . . . **they forget, or just require a little more supervision compared to before**."*

*(Focus Group 1, Fitness Supervisor)*

The formal and informal processes to identify cognitive impairment among program participants was thought to be important for tailoring exercise programs and delivery. This monitoring of abilities by exercise providers formed the basis for continual adaptation of exercises to support inclusion as well as progression, elaborated in Sub-theme 2b below.

**Sub-theme 2b: Exercise providers develop ad-hoc strategies to overcome barriers and support and exercise for people living with dementia.** Exercise professionals described strategies to support exercise among persons with MCI or dementia in three areas: 1) purposeful selection of exercises; 2) adapted exercise delivery; and 3) purposeful progression or regression of difficulty.

Exercise providers in this study purposely selected exercises they considered safe and easy to execute, as well those that would provide and provide functional benefits to persons with MCI or dementia. Enhanced safety was both a consideration in selecting appropriate exercises as well as a target outcome of exercises recommended. In terms of selecting safe exercises, for example, exercise providers avoided more cognitively demanding exercises that increased the risk of falling (e.g. treadmill walking), opting instead to modify exercises by using safer equipment (e.g. recumbent biking, track walking). Another example of these safety modifications made by exercise providers was promoting the use of weight machines with guided movement patterns to reduce the potential for injury during resistance training instead of free weights.

Exercise providers identified many benefits to participating in exercise for clients with MCI or dementia, which guided their exercise selection for these clients. Exercise providers described selecting activities with clear functional benefits, such as those which are known to improve balance, reduce falls risk, and enhance performance of daily life activities for their clients with MCI or dementia. Exercise providers participating in this study recognized that aerobic exercise could promote brain and vascular health, while improving cardiovascular fitness and managing comorbid health conditions. However, they were uncertain about the cognitive benefits of exercise, but still believed that cognitively stimulating exercises were preferable to purely physical activities so that they could engage both mind and body.

Exercise providers also recognized that their exercise delivery should be tailored to be more inclusive for persons with MCI or dementia. While they did not know of any established 'best practice' for tailoring exercise, exercise providers discussed several pragmatic changes to exercise delivery that they developed on an ad hoc basis to meet the needs of their clients with dementia outlined in Table 2.

**Table 2. Strategies used by exercise providers to accommodate individual-level barriers to exercise among persons with MCI or dementia.**

| Perceived Source of Challenge | Challenge to Exercise Delivery | Potential Solution or Support |
|---|---|---|
| Memory Impairment | Need for supervision for safety | Trained volunteers to support participant |
| | Difficulty remembering (more time to learn exercises, difficulty remembering exercise techniques) | Auditory, visual, and physical cueing to correct movement patterns |
| Decreased attention | Deviations for exercise program, wandering | Trailed volunteer to support participant |
| | | Include support person in the exercise program |
| Movement & gait impairments | Need for accessible exercise equipment and ability to adapt exercise program with safe and appropriate exercises | Use of auditory, visual, and physical cueing to help persons with dementia follow correct movement patterns |
| | | Adapt exercises to physical capabilities (e.g. machines with guided movement patterns instead of free-weights, cycling instead of treadmill walking) |
| Sensitivity to over-stimulating environments | Modify or choose environment to allow for reduced visual distractors, noisiness | Adapt environment to reduce sensory stimulation (e.g., quiet, familiar music when desired) |
| Stigma & lack of familiarity with exercise environment | Need to actively make the persons with dementia feel comfortable in the exercise program | Progress from individual exercise to group exercise |
| | | More support early in the program |
| | | Adapt programming according to changing needs |
| Emotional reactivity and agitation | Behavioural or emotional outbursts | Observe carefully to recognize behavioural and emotional triggers |
| | | Anticipate these triggers and adapt |
| | | Adapting program daily depending on the emotional state of the participant |
| Wandering | Need for increased supervision | Trailed volunteer to support participant |
| | | Include support person in the exercise program |

Finally, participants also discussed using a staged approach to programming as one method to adapt exercise delivery to be more inclusive of persons with MCI or dementia. Exercise providers progressed or regressed the difficulty of clients' programs based on fluctuations in physical and cognitive function in the short- and long-term. Specifically, exercise providers discussed progressing independence in exercise, where participants receive more support when starting the program before progressing to more independent exercise. That is, as participants experience initial gains in physical function, exercise competency, and self-confidence, programs could progress to lower supervision and greater complexity. However, exercise difficulty could also be decreased based on acute or long-term declines in function–that is, on 'bad days' or as cognitive or physical impairment progresses. One participant in particular described specific strategies to adapt level of independence in exercise programs including care partners of persons with MCI or dementia:

*"And I think when people start, they get a little more individualized attention, and then after a few times they might be able to move to more general program... **So it's kind of like a starter program, and then like a maintenance program** ... like, gradual step in to build on what they learn, because I feel like the more individualized attention they get initially the better...**They [caregivers] could stay if they wanted to stay and watch or participate [in exercises] with the**m. Or even initially, like sometimes with small kids and swimming lessons the parents go in the pool for like the first four weeks, and then the second four weeks they come out of the pool and the kids go in by themselves... something similar to that where they're kind of there, for the first couple and then once they're established, they can attend on their own."*

*(Focus Group 1, Fitness Supervisor)*

Despite a lack of established best practice, exercise providers described several ways they adapt exercise program content and delivery for persons living with MCI dementia. This included a recognition, by some, that the type and level of support provided would vary over time.

## Theme 3—Training and best practices, with flexibility

The third theme describes exercise providers' needs for education and resources so they can better accommodate persons with MCI or dementia, as little training currently exists. Broadly, exercise professionals expressed a willingness during focus groups to include and accommodate persons with MCI or dementia within their programs, and several were already doing so. However, they also perceived a lack of dementia-specific education and training resources for exercise providers. Few participants had any formal training regarding dementia-inclusive exercise. They highlighted that continuing education opportunities exist for other specific health conditions (e.g. osteoporosis, heart failure), but do not currently exist for MCI or dementia.

> *"In terms of like workshops and stuff, I don't know . . .I'm like a member of different things and receive like newsletters from certain groups. . .. we've gone to like a workshop for exercise with people that have osteoporosis . . . I've gone to like a heart health one, but **I've never really come across anything like that for people with dementia though**."*

> (*Focus Group 1, Fitness Supervisor*)

Participants also noted a need to develop consensus around best practices—that is, approaches and strategies for exercise programming and delivery with demonstrated efficacy, for administering exercise and program design for persons with MCI and dementia. They recognized, however, that any best practices would still need to be tailored to the resources available to the exercise provider and their organization, as well as the abilities and preferences of their clients with MCI or dementia. For example, best practices may have to be adapted based on factors such as the location of exercise (centre- vs. home-based), care partner involvement (involved or not), program size (individual vs. group), and group composition (dementia-specific vs. -integrated).

> *"And I think it would be good to look at. . . if there are any other programs . . .like that. Like, if they exist and what they're like and how they've found success. Like, would you want to do . . . more of like a group exercise, um, which I see them -they have small groups but they're all doing exercises out in the gym together, or do you wanna . . .would you do it like our program like have-everyone have their own individual program? So you'd have to kinda look . . . I mean you could just try it and see what works best but you'd wanna look at . . .how people have done it in the past."*

> (*Focus Group 1, Fitness Supervisor*)

In the absence of formal educational and professional resources and guides, exercise professionals described engaging in *ad hoc* information-seeking processes to meet the needs of their clients with cognitive impairment. Sometimes, this involved consulting with other health care providers or groups that support dementia research and services. Furthermore, some exercise providers considered persons with dementia to be beyond their scope of practice (which is just exercise provision), in which case they referred clients to outside services more appropriate to their needs.

*"Well, if it's not within our scope of practice we usually try and find a place to refer them to. So our scope of practice is just the "at exercise" aspect . . ."I've connected with someone from the University before. . . because **I wasn't really sure on things to do or ways to help improve cognitive function while they're exercising**. We do have, like, people within the university that we can connect with to get more information."*

(*Focus Group 1, Fitness Supervisor*)

Due to this lack of formal educational and professional resources, exercise professionals participating in this study voiced that they learned about dementia as needed when individuals with MCI or dementia came to them for exercise. Though this individual accommodation is partially a result of lack of training regarding dementia, participants also recognized that some adaptation is likely to be required regardless of training due to the heterogeneity of MCI and dementia experiences, symptoms, abilities, and needs.

*"Often times when you hear a chronic condition, you kind of lump people into that category. Especially in a diagnosis of MCI or Alzheimer's, there is such a range of impairment. . . **so it** [exercise] **ideally needs to be somewhat individualized to accommodate those people**. . . Because there's-there's just such a range of differences within folks that have that . . . **you want to challenge appropriately and not assume** [what] **they are only capable of**."*

(*Focus Group 3, Registered Kinesiologist*)

## Discussion

This study explored the perceptions, experiences, and needs of community exercise providers for delivering exercise to persons with MCI or dementia. Exercise providers' personal and professional experiences with MCI and dementia shape their views of persons with MCI or dementia, and their capacity to participate in exercise. Exercise providers described modifying exercise delivery for persons with MCI or dementia on an *ad hoc* basis but were able to identify strategies to accommodate these groups. Our results highlight a need for training and education regarding exercise for persons living with MCI or dementia to promote a consistent understanding of dementia and provision of inclusive services, while still recognizing the ongoing need to adapt on a case-by-case basis due to the heterogeneity of dementia.

The heterogeneity of dementia, by symptoms and severity, supports the need for dementia-training among a broad range of exercise providers. Dementia-specific exercise programs tend to focus on those with more extensive cognitive or physical needs (e.g. day programs). Those who are young or physically and cognitively fit enough to participate in more challenging programs are often left out or forced to take up exercise programs meant for the general older adult population, where exercise providers may have little understanding of dementia-inclusive practices. Having more exercise providers trained to deliver dementia-inclusive programs could increase the portion of opportunities for persons living with MCI or dementia to participate in recreation, leisure, and sport, as is their right per the United Nations Convention on the Rights of Persons with Disabilities [24]. As a result, designing and delivering exercise programs with dementia-inclusive principles is both a need and a right in many countries. The Centres for Disease Control (United States) and the National Centre on Physical Activity, Health, and Disabilities (United States) have resources available to support the design of accessible health promotion programming, but their materials are not specific to cognitively-impaired older adults [36, 37].

This study builds on two prior studies in order to better understand exercise delivery and implementation for persons with MCI or dementia [30, 31], and is the first to specifically focus on community exercise providers rather than healthcare professionals. Our results suggest that exercise providers' beliefs about the capacity of persons with MCI or dementia to engage in exercise are closely linked to personal and professional experiences and not to education and training, which was generally lacking. Although all the exercise providers interviewed had some personal or professional experience with persons with MCI or dementia, the frequency and nature of experiences with dementia were highly variable. Not surprisingly, the ability of exercise providers to recognize the variability of dementia and associated symptoms was more developed among exercise providers who had engaged with many persons living with dementia. Systematic education regarding the stages and symptoms of dementia and variability between individuals would create a more accurate and consistent understanding of the broad range of abilities, experiences, and needs of persons with MCI or dementia in exercise settings.

In this study, exercise providers' perceptions of dementia focused on cognitive and behavioural symptoms, in line with diagnostic criteria that focus on these changes [38]. However, they did not often identify sensory and motor changes that are common with MCI or dementia [39, 40], which would be important to recognize and accommodate in exercise delivery. For example, visual hallucinations are one of the core diagnostic criteria for dementia with Lewy bodies [41], so persons living with this form of dementia may require specific environmental or training style adaptations to reorient them to the present setting. In addition, poor balance and falls are more common in persons living with dementia of many types, as compared to cognitively healthy older adults [40, 42] and may influence appropriate selection of exercise mode.

Our finding that exercise providers used *ad hoc* strategies to recognize and accommodate persons with MCI or dementia is consistent with a prior study of physiotherapists' experiences administering a high-intensity functional exercise program in residential care settings in Sweden [30]. The physiotherapists in the study described developing strategies for exercise delivery through an iterative approach, which led to a larger repertoire of strategies to be used with persons with dementia over time. The findings of this prior study emphasized the iterative process of working with clients who have cognitive impairment to identify training solutions that work for their unique abilities and preferences [30]. Perhaps due more variability in the level of engagement with persons with MCI or dementia, accumulation of knowledge was not a theme that was developed based on interviews with our participants. Community exercise providers who engage regularly with persons with MCI or dementia may also accumulate better skills. Future efforts should seek to crystallize the knowledge and experiences of exercise providers in order to include this expert knowledge as part of education and training, which would ensure a common starting point for practice and less reliance on individual efforts for quality control. In particular, formalized education could increase the breadth and depth of understanding regarding the symptoms and experiences of persons with MCI and dementia. Integration of MCI and dementia into formal exercise training modules could have significant social impact by reducing the stigma around dementia and improving exercise program inclusivity.

Both the prior study of physiotherapists and the current study highlight the need to adapt exercise delivery to the individual due to the heterogeneity of dementia [30, 43]. Even with best-practice education and training, it is likely that exercise providers would need to adapt delivery to the individual, as well as to fluctuations in the individual's abilities day to day and over time. Peer-facilitated knowledge exchange, on-line or in-person, could complement formal training programs and provide a forum to address novel challenges and identify appropriate adaptations for the individual.

## Study limitations

While this study employed a robust methodology to support the rigor of its findings, there are some unavoidable limitations. The regional specificity and the predominance of women in this study limits the direct transfer of its findings to other regions and settings and may not well reflect the experiences and views of male exercise providers. Recruitment was targeted specifically to community exercise providers who had experience working with older adult clients, with or without cognitive impairment. The experiences, perceptions, and needs of these exercise providers is likely different from exercise providers who do not work with older adults. Furthermore, the participants in this study may be more interested in exercise delivery among those with MCI or dementia relative to other exercise providers. As a result, their perceptions of exercise may differ from average. However, given the specific experiences of participants in this study around dementia-friendly exercise, it is likely that they were more knowledgeable about this topic and better equipped to make specific, practical recommendations to increase accessibility of community exercise to persons with dementia.

## Conclusion

Persons with MCI or dementia have a right to participate in exercise programs that support their health, well-being, and functional abilities. Community exercise providers who train older adult clients are in a unique position to enable dementia-inclusive exercise opportunities. Perceptions of exercise providers about dementia are shaped by their variable and limited experiences of dementia in personal and professional settings. In the absence of formal training and best practices on dementia, strategies to adapt exercise are largely symptom-focused and implemented on a case-by-case basis. Exercise providers have identified a need for formal education and training around dementia-inclusive exercise but recognize that there will always be a need to adapt due to the heterogeneity of dementia. Future research and initiatives should develop evidence-based training for exercise providers and evaluate knowledge translation and implementation strategies to understand the effectiveness of various approaches in creating dementia-inclusive exercise opportunities.

## Supporting information

**S1 File. Focus group discussion guide.**
(DOCX)

## Acknowledgments

The Authors would like to acknowledge the Dementia-Inclusive Choices for Exercise research team, whose contributions continue to shape related work, and the contributions of Nic Hobson during data collection.

## Author Contributions

**Conceptualization:** Cheyenne Mitchell, Kayla Regan, Maximillian Bergelt, Sherry Dupuis, Lora Giangregorio, Laura E. Middleton.

**Data curation:** Lauren E. Bechard, Cheyenne Mitchell, Kayla Regan, Maximillian Bergelt, Laura E. Middleton.

**Formal analysis:** Lauren E. Bechard, Aidan McDougall, Cheyenne Mitchell, Maximillian Bergelt, Laura E. Middleton.

**Funding acquisition:** Aidan McDougall, Cheyenne Mitchell, Laura E. Middleton.

**Investigation:** Lauren E. Bechard, Cheyenne Mitchell, Laura E. Middleton.

**Methodology:** Cheyenne Mitchell, Kayla Regan, Maximillian Bergelt, Sherry Dupuis, Lora Giangregorio, Laura E. Middleton.

**Project administration:** Cheyenne Mitchell, Kayla Regan, Laura E. Middleton.

**Resources:** Laura E. Middleton.

**Software:** Laura E. Middleton.

**Supervision:** Laura E. Middleton.

**Writing – original draft:** Lauren E. Bechard, Aidan McDougall, Kayla Regan, Laura E. Middleton.

**Writing – review & editing:** Lauren E. Bechard, Aidan McDougall, Cheyenne Mitchell, Kayla Regan, Maximillian Bergelt, Sherry Dupuis, Lora Giangregorio, Shannon Freeman, Laura E. Middleton.

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
