## [Decision Letter · Decision Letter 0]

26 Jun 2020

PONE-D-20-09999

Dementia- and Mild Cognitive Impairment-Inclusive Exercise: Perceptions, Experiences, and Needs of Community Exercise Providers.

PLOS ONE

Dear Dr. Bechard,

Thank you for submitting your manuscript to PLOS ONE. After careful consideration, we feel that it has merit but does not fully meet PLOS ONE’s publication criteria as it currently stands. Therefore, we invite you to submit a revised version of the manuscript that addresses the points raised during the review process.

We look forward to receiving your revised manuscript.

Kind regards,

Coralie English, PhD

Academic Editor

PLOS ONE

Journal Requirements:

Additional Editor Comments:

PONE-D-20-09999

Dementia and exercise qual

Editor comments

Congratulations on a well written paper. Reviewer 1 has raised some important points about the description of the methods in particular that require addressing. I also have a few minor comments – see below. I look forward to seeing a revised submission

Abstract

• Use of diverse twice and disparate once in the one sentence – can this be reworded to be more descriptive?

Methods

• Clearly written – please consider reviewer 1’s comments on the description of the methodology and the recruitment process

Results

• Table 1 – please remove decimals for age and percentages and provide mean and SD for age rather than median. Inter-quartile range instead of, or in addition to range should be provided for median scores. The column heading and results should be n (%), not the other way around – ie rural practice 1 (3%).

• Please rename the organisation type in Table 1 to be more descriptive. Eg not-for-profit community organisations’ and ‘for profit fitness centres’ as described later in text are more descriptive. What are ‘independent’ organisations?

• Please reword ‘exercise designations’ to ‘profession’

• Line 222 reword to ….” Exercise providers who reported extensive experiences with MCI and dementia in professional settings tended more to were more likely to recognize the diversity of abilities and experiences of persons with MCI or dementia.

• Table 2 with clear suggestions for practical strategies is great

• Flow is logical and writing is clear

Discussion

• Well written and covers relevant and appropriate points

Reviewers' comments:

Reviewer's Responses to Questions

**Comments to the Author**

1. Is the manuscript technically sound, and do the data support the conclusions?

Reviewer #1: Yes

2. Has the statistical analysis been performed appropriately and rigorously? 

Reviewer #1: N/A

3. Have the authors made all data underlying the findings in their manuscript fully available?

Reviewer #1: No

4. Is the manuscript presented in an intelligible fashion and written in standard English?

Reviewer #1: Yes

5. Review Comments to the Author

Reviewer #1: This paper presents research addressing an important gap – the experiences of community exercise providers in providing exercise opportunities for people with dementia or cognitive impairment. The authors are to be congratulated on an elegantly written paper with important practical application.

A few suggestions are made below. One is of particular importance, relating to the use of thematic analysis.

Introduction

– an exceptionally well-written introduction that takes the reader through a clear process of thought to ‘land on’ the aims of the study. The opening paragraph speaks of the economic cost, which is clearly important, but I wonder if the impact of dementia on individuals could also be acknowledged.

Methods

– Recruitment (line 116) – please clarify what you mean by ‘outreach’. Did you ‘cold call’ organisations? Did they then identify participants, or did they distribute materials to their staff? Did you purposively sample organisations you already had a relationship with?

– Data analysis (line 149 onwards) – can you please describe why you chose a semantic approach and perhaps also include a brief description of thematic analysis?

– Consensus coding structure (line 159) – can you please describe why this approach was chosen or provide a citation to support this approach? I’m not sure this is part of Braun and Clarke’s approach to thematic analysis?

Results

– Three themes emerged (line 203) – some qualitative researchers, including Braun and Clarke suggest that themes do not ‘emerge’. As the authors have chosen Braun and Clarke’s approach to thematic analysis may I suggest this wording is reconsidered? Alternatives may include themes being ‘generated’ or ‘developed’.

– I am not sure if thematic analysis is the best description of the approach taken. The themes seem to follow the questions posed in the question guide provide in the Appendix. The themes may be better described as domain summaries. The information presented is very interesting and useful – Table 2 is particularly helpful. Therefore, it would seem to me that the authors have three options (although the authors may have another idea):

1. To more overtly describe how a thematic analysis approach was applied so the reader can better understand

2. To conduct further analysis and present fully realised themes

3. Consider reporting a different approach. Qualitative description may possibly be an alternative?

Discussion

– Well written. This may need tweaking depending on changes made to the results reporting.

References

– Year of publication is missing for each reference and URLs are not included for online materials.

6. PLOS authors have the option to publish the peer review history of their article (what does this mean?). If published, this will include your full peer review and any attached files.

Reviewer #1: No

---

## [Author Response · Author response to Decision Letter 0]

23 Jul 2020

Response to Reviewers – Manuscript PONE-D-20-0999

Journal Requirements:

1. Please ensure that your manuscript meets PLOS ONE’s style requirements, including those for file naming. The PLOS ONE style templates can be found at: https://journals.plos.org/plosone/s/file?id=wjVg/PLOSOne_formatting_sample_main_body.pdf and https://journals.plos.org/plosone/s/file?id=ba62/PLOSOne_formatting_sample_title_authors_affiliations.pdf

• Main body has been reformatted to guidelines

• Title page has been reformatted to guidelines.

• ORCID iD has been added for the corresponding author.

• Supporting information file has been captioned at the end of the manuscript and amended for PLOS ONE file naming convention.

Additional Editor Comments:

Editor comments

Abstract

Use of diverse twice and disparate once in the one sentence – can this be reworded to be more descriptive?

• This has been reworded to be more descriptive.

Methods

Clearly written – please consider reviewer 1’s comments on the description of the methodology and the recruitment process

• See response to reviewer 1’s comments on methodology description and recruitment process.

Results

Table 1 – please remove decimals for age and percentages and provide mean and SD for age rather than median. Inter-quartile range instead of, or in addition to range should be provided for median scores. The column heading and results should be n (%), not the other way around – ie rural practice 1 (3%).

• Table 1 values have been changed to means with the range provided and column heading to n (%). Decimals have been removed for the age value. Professional designations are not mutually exclusive (participants could and did hold more than one type of designation). A note reflecting this has been added following table 1 to provide context for interpretation of percentages in 

Please rename the organisation type in Table 1 to be more descriptive. Eg not-for-profit community organisations’ and ‘for profit fitness centres’ as described later in text are more descriptive. What are ‘independent’ organisations?

• Organization types have been renamed to be more descriptive in table 1. “Independent” refers exercise providers who are not affiliated with specific centres (e.g. independent contractors). Corresponding wording in the results section has been amended for consistency with Table 1. 

Please reword ‘exercise designations’ to ‘profession’

• The exercise designations have been re-worded as “professional designations” for clarity in Table 1 and reworded for consistency in the results section.

Line 222 reword to ….” Exercise providers who reported extensive experiences with MCI and dementia in professional settings tended more to were more likely to recognize the diversity of abilities and experiences of persons with MCI or dementia.

• I’m unclear on what you are requesting in this wording change, but I specifically chose the phrasing “tended more to” because “were more likely to” implies statistical calculation of likelihood, which was not done in this qualitative study.

Table 2 with clear suggestions for practical strategies is great

Flow is logical and writing is clear

• Thank-you!

Discussion

Well written and covers relevant and appropriate points

• Thank-you!

Review Comments to the Author:

Reviewer #1: This paper presents research addressing an important gap – the experiences of community exercise providers in providing exercise opportunities for people with dementia or cognitive impairment. The authors are to be congratulated on an elegantly written paper with important practical application.

A few suggestions are made below. One is of particular importance, relating to the use of thematic analysis.

Introduction

– an exceptionally well-written introduction that takes the reader through a clear process of thought to ‘land on’ the aims of the study. The opening paragraph speaks of the economic cost, which is clearly important, but I wonder if the impact of dementia on individuals could also be acknowledged.

• A statement on the broad impact of dementia has been added with a supporting reference.

Methods

– Recruitment (line 116) – please clarify what you mean by ‘outreach’. Did you ‘cold call’ organisations? Did they then identify participants, or did they distribute materials to their staff? Did you purposively sample organisations you already had a relationship with?

• This comment has been addressed in the materials & methods section.

– Data analysis (line 149 onwards) – can you please describe why you chose a semantic approach and perhaps also include a brief description of thematic analysis?

• A description of this has been provided in the first paragraph of the Data Analysis section, but this description has been reworded to enhance clarity on why a semantic approach was chosen over a latent approach. An enhanced description of the thematic analysis method was added, as per the reviewer’s later comment on the results section.

– Consensus coding structure (line 159) – can you please describe why this approach was chosen or provide a citation to support this approach? I’m not sure this is part of Braun and Clarke’s approach to thematic analysis?

• Consensus coding was used to reduce the potential for influence of an individual researcher’s bias on data analysis and enhance the confirmability of results. A statement on this and supporting citation have been added.

Results

– Three themes emerged (line 203) – some qualitative researchers, including Braun and Clarke suggest that themes do not ‘emerge’. As the authors have chosen Braun and Clarke’s approach to thematic analysis may I suggest this wording is reconsidered? Alternatives may include themes being ‘generated’ or ‘developed’.

• Thank-you for this suggestion about rewording. The phrasing around “emerge” has been changed to reflect this suggestion (Line 424). Wording relating to the concept of themes “emerging” has been removed from other areas of the manuscript (abstract, results, discussion).

– I am not sure if thematic analysis is the best description of the approach taken. The themes seem to follow the questions posed in the question guide provide in the Appendix. The themes may be better described as domain summaries. The information presented is very interesting and useful – Table 2 is particularly helpful. Therefore, it would seem to me that the authors have three options (although the authors may have another idea):

1. To more overtly describe how a thematic analysis approach was applied so the reader can better understand

2. To conduct further analysis and present fully realised themes

3. Consider reporting a different approach. Qualitative description may possibly be an alternative?

• Thank-you for this thoughtful feedback. The authors agree that the findings do, to an extent, reflect the questions in the guide provided in the appendix, but this is because an inductive, semantic approach to analysis was used. Findings yielded by this approach will naturally be reflective of the questions asked in the interview guide – analysis was driven by ideas and statements provided by participants in response to questions posed in the interview guide, and we did not seek to infer underlying meanings of participants experiences, perceptions, and needs. The themes reported in the results section, while related to the domains in the interview guides, do not directly map on to the domains of questions in the interview guide. For example, Theme 1 is based on data about the relationship between experiences and perceptions of exercise providers, and is not simply a statement about one or the other. In another example of distinction between question guide and themes reported, there is a specific question about home-based exercise programs, but this was not a broad theme reflected in the data pertaining to the experiences, perceptions, and needs of exercise providers. A more thorough description of thematic analysis was provided to help readers better understand how the researchers arrived at the results of this study (option 1, as suggested by the reviewer).

Discussion

– Well written. This may need tweaking depending on changes made to the results reporting.

• Thank-you. No changes have been made to discussion as the reporting of results were not changed.

References

– Year of publication is missing for each reference and URLs are not included for online materials.

• References have been updated to include year, URL, and DOI where missing.

---

## [Editor Report · Decision Letter 1]

5 Aug 2020

PONE-D-20-09999R1

Dementia- and Mild Cognitive Impairment-Inclusive Exercise: Perceptions, Experiences, and Needs of Community Exercise Providers.

PLOS ONE

Dear Dr. Bechard,

Thank you for submitting your manuscript to PLOS ONE. After careful consideration, we feel that it has merit but does not fully meet PLOS ONE’s publication criteria as it currently stands. Therefore, we invite you to submit a revised version of the manuscript that addresses the points raised during the review process.

We look forward to receiving your revised manuscript.

Kind regards,

Coralie English, PhD

Academic Editor

PLOS ONE

Additional Editor Comments (if provided):

Thank you for your prompt response to all the issues raise. I have only a couple of outstanding revisions before I am happy to accept this paper:

• Line 216 and elsewhere – please change ‘professional designations’ to ‘professional qualifications’ or ‘professional backgrounds’

• Remove “ “ from “other”

• Please reword sentence now on line 241 from ‘tended more to’ to “were more likely to” – ‘tended more to’ is very clunky phrasing and in the context of a qual study ‘more likely to’ will not be misinterpreted as inferring statistical likelihood

---

## [Author Response · Author response to Decision Letter 1]

5 Aug 2020

Response to Reviewers – Manuscript PONE-D-20-0999

Thank-you for your prompt response to the revisions that were submitted. Responses to the editor’s comments are provided below and are reflected in the updated manuscript submissions.

Editor’s comments

Line 216 and elsewhere – please change ‘professional designations’ to ‘professional qualifications’ or ‘professional backgrounds’

• This has been changed to qualifications / backgrounds as appropriate in Line 216 and elsewhere.

Remove “ “ from “other”

• This has been removed in Line 217.

Please reword sentence now on line 241 from ‘tended more to’ to “were more likely to” – ‘tended more to’ is very clunky phrasing and in the context of a qual study ‘more likely to’ will not be misinterpreted as inferring statistical likelihood

• This has been reworded in Line 241.

---

## [Editor Report · Decision Letter 2]

12 Aug 2020

Dementia- and Mild Cognitive Impairment-Inclusive Exercise: Perceptions, Experiences, and Needs of Community Exercise Providers.

PONE-D-20-09999R2

Dear Dr. Bechard,

We’re pleased to inform you that your manuscript has been judged scientifically suitable for publication and will be formally accepted for publication once it meets all outstanding technical requirements.

Kind regards,

Coralie English, PhD

Academic Editor

PLOS ONE
---

## [Editor Report · Acceptance letter]

28 Aug 2020

PONE-D-20-09999R2 

Dementia- and Mild Cognitive Impairment-Inclusive Exercise: Perceptions, Experiences, and Needs of Community Exercise Providers. 

Dear Dr. Bechard:

I’m pleased to inform you that your manuscript has been deemed suitable for publication in PLOS ONE. Congratulations! Your manuscript is now with our production department. 

If your institution or institutions have a press office, please let them know about your upcoming paper now to help maximize its impact. If they’ll be preparing press materials, please inform our press team within the next 48 hours. Your manuscript will remain under strict press embargo until 2 pm Eastern Time on the date of publication. For more information please contact onepress@plos.org.

Kind regards, 

on behalf of

Professor Coralie English 

Academic Editor

PLOS ONE